# Detecting Geometric Deformation in Visual Generation

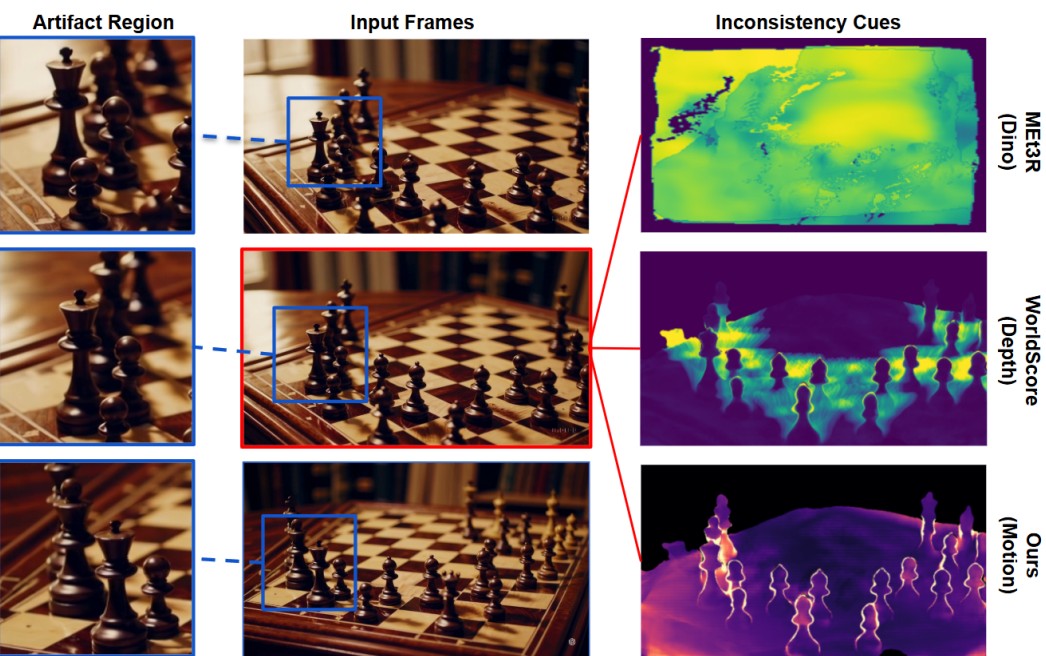

Figure 1: **Artifact (geometric deformation) detection on a generated video.** (**Top**) MEt3R produces a diffuse error map and fails to localize the specific geometric error, as it relies on semantic feature (DINO) consistency. (**Middle**) WorldScore (Depth Reprojection) correctly identifies the inconsistent object by evaluating 3D depth consistency, but its resulting map is not sharply localized. (**Bottom**) **Our approach** uses motion cues to isolate non-rigid flow and produces a sparse and interpretable map that precisely pinpoints the subtle deformation.

## Abstract

Recent text-to-video and multi-view generative models produce striking imagery but often violate basic 3D geometry, exhibiting non-rigid "melting" or "breathing" artifacts across viewpoints. We study this failure mode in the static-scene regime, where camera motion is allowed but objects must remain rigid; any apparent object motion is deemed deformation. We introduce a geometry-grounded detection pipeline that localizes and quantifies such artifacts. The pipeline estimates camera motion and depth to predict the rigid pixel motion expected in a static world, compares it to observed optical flow to obtain a motion error map, and fuses this with a depth reprojection error map to handle occlusions. The result is an occlusion-aware, per-pixel deformation map and interpretable video-level scores. To enable controlled, quantitative evaluation, we present WARPBENCH, a synthetic dataset that applies localized thin-plate-spline warps to real frames while recording dense displacement ground truth. We instantiate it as *CO3D-Warp* (object-centric) and *ScanNet++-Warp* (scene-level). To probe performance

beyond synthetic perturbations, we further introduce **Geo-Flaw**, a task-oriented benchmark spanning object-centric reconstruction, indoor navigation, large-scale outdoor scenes, and challenging surfaces, under both slow and fast camera motion. Our experiments show that the proposed pipeline detects deformation artifacts missed by feature-based metrics and coordinate-only consistency measures, and it naturally extends to moving object segmentation, outperforming prior training-free baselines. Together, these components provide an interpretable and practical toolkit for diagnosing geometric inconsistency and for benchmarking video generative models on true 3D fidelity.

# 1 INTRODUCTION

Generative models for multi-view imagery and novel view synthesis (Yu et al., 2023b; Seo et al., 2024; Rombach et al., 2021) have advanced rapidly, producing photorealistic frames from text prompts or a single image. Yet, despite impressive visual quality, the generated images by these models frequently violate basic 3D geometry: objects stretch, bend, or melt across viewpoints, revealing deformation artifacts that are inconsistent with a rigid scene. To investigate this issue, this work explicitly targets *static videos with no moving objects*; any apparent object motion is treated as deformation. In this setting, the camera moves while scene geometry should remain unchanged, but generated views often exhibit structural drift between frames.

Evaluating multi-view (3D) consistency using this simplified setting is challenging for existing methods. Prior approaches either compare deep features across warped views (for example, DINO-based metrics such as MEt3R (Asim et al., 2025)) or rely on depth- and point-cloud–based errors (for example, WorldScore-3D consistency). Feature comparisons can capture semantic drift but are intentionally insensitive to local shape changes and often miss geometric deformation. Depth- and point-cloud–based errors are sensitive only to 3D coordinates or depth; if corresponding points occupy similar 3D locations after alignment, these methods can report low error even when surfaces have bent, sheared, or otherwise deformed. They also provide limited diagnostic insight into *where* and *how* rigidity is violated.

We propose a geometry-grounded *detection pipeline* that measures deformation directly from motion cues and depth reprojection. We estimate camera motion and scene depth, compute the rigid pixel motion that would occur if the scene were perfectly static, and compare it to the optical flow observed between generated frames to obtain a motion error map. Because the motion error map is unreliable in occluded regions, we complement it with a depth reprojection error map. Fusing these two signals yields an occlusion-aware, per-pixel *deformation map* that localizes violations of rigidity and can be aggregated into interpretable video-level scores.

Annotating artifact ground truth on generated videos is difficult. To enable quantitative assessment with reliable supervision, we introduce WARPBENCH, a synthetic deformation dataset that applies localized, non-rigid thin-plate-spline warps to real frames while recording the exact displacement used to distort each image. We instantiate WARPBENCH on object-centric clips from **CO3D** (Reizenstein et al., 2021) (*CO3D-Warp*) and scene-level reconstructions from **ScanNet++** (Dai et al., 2017; Yeshwanth et al., 2023) (*ScanNet++-Warp*). Each instance provides a dense per-pixel displacement field and an occlusion indicator, enabling precise, per-pixel evaluation of detectors as well as scalar summaries via displacement magnitude. We quantitatively demonstrate the effectiveness of our pipeline on WARPBENCH, and then use it to benchmark state-of-the-art video generation models in terms of deformation artifacts.

To probe performance beyond synthetic warps, we introduce **Geo-Flaw**, a task-oriented benchmark for static scenes that spans four scenario families, including object-centric reconstruction, indoor navigation, large-scale outdoor reconstruction, and challenging surfaces and edges. Besides, the benchmark covers both slow and fast camera motions: slow motions reveal subtle "breathing" or "melting" artifacts, and fast motions challenge multi-view coherence under aggressive perspective changes. This design supports a structured evaluation of a model's ability to maintain a stable and plausible 3D world.

We make the following contributions:

- A novel pipeline that fuses residual motion and depth reprojection errors into interpretable, dense deformation maps.

- **WARPBENCH**, a synthetic dataset with dense ground-truth warps (*CO3D-Warp* & *ScanNet++-Warp*) for rigorous evaluation.

- **Geo-Flaw**, a comprehensive benchmark and analysis of geometric artifacts in leading text-to-video and multi-view generation models.

These components provide a principled toolkit for diagnosing and measuring geometric inconsistency in generated videos of static scenes.

## 2 RELATED WORK

Our work positions at the intersection of generative model evaluation, 3D computer vision, and motion analysis. We situate our contributions with respect to prior work in evaluating multi-view consistency and in the foundational tasks of motion decomposition and occlusion handling.

### 2.1 METRICS FOR MULTI-VIEW GEOMETRIC CONSISTENCY

Evaluating the 3D consistency of generative models is an active area of research, with methods largely falling into two categories: feature-based and coordinate-based.

**Feature-Based Consistency.** A popular approach is to measure the semantic similarity between views. MEt3R (Asim et al., 2025), for instance, computes the cosine similarity of dense DINO (Caron et al., 2021) features between a rendered view and a source view warped by predicted depth and camera motion. While effective for capturing large-scale semantic drift, this approach has a fundamental limitation for our task: deep features are often designed to be invariant to the very local geometric deformations we aim to detect. A pillar that is slightly bent might be geometrically incorrect but semantically identical to a straight one, leading feature-based metrics to miss such artifacts. Our method, in contrast, moves away from feature similarity and instead focuses on the coherence of motion fields to directly target these subtle structural inconsistencies.

**Coordinate- and Depth-Based Consistency.** Another line of work evaluates consistency by measuring errors in the 3D positions of points. For example, the Thresholded Symmetric Epipolar Distance (TSED) (Yu et al., 2023a) measures consistency based on the epipolar geometry of sparse SIFT (Lowe, 2004) feature matches. More recent methods like WorldScore (Duan et al., 2025) compute a scalar reprojection error after performing a full structure-from-motion (SfM) reconstruction. Similarly, MVGBench (Xie et al., 2025) evaluates object-centric models by sampling point clouds from generated views and comparing them using the Chamfer distance.

A key drawback of these methods is that they typically output a single numerical score for an entire video or set of views. This lacks diagnostic power, as it does not reveal *where* or *how* the scene geometry fails. Furthermore, by focusing only on point positions or depth, they can fail to detect surface-level distortions like bending or shearing if the aligned 3D coordinates remain close. Our pipeline addresses this gap by producing a dense, per-pixel deformation map, offering interpretable, localized feedback on geometric violations.

### 2.2 MOTION DECOMPOSITION AND SEGMENTATION

Our pipeline's core idea—disentangling camera ego-motion from independent object motion—is a classic problem in computer vision. We build upon powerful, modern optical flow models like UFM (Zhang et al., 2025) to estimate the dense pixel correspondence between frames. By subtracting the predicted rigid flow, we isolate a residual motion field that corresponds to non-rigid deformation.

This formulation naturally connects our work to moving object segmentation. While many methods exist for this task, our approach is notable for being training-free. Methods like Segment Any Motion in Videos (Huang et al., 2025) represent the state-of-the-art but require training. By treating any non-rigid motion as "foreground," our deformation map serves as a powerful signal for segmentation. We

demonstrate that this simple, geometry-grounded approach outperforms other training-free baselines on the DAVIS dataset (Perazzi et al., 2016; Pont-Tuset et al., 2017).

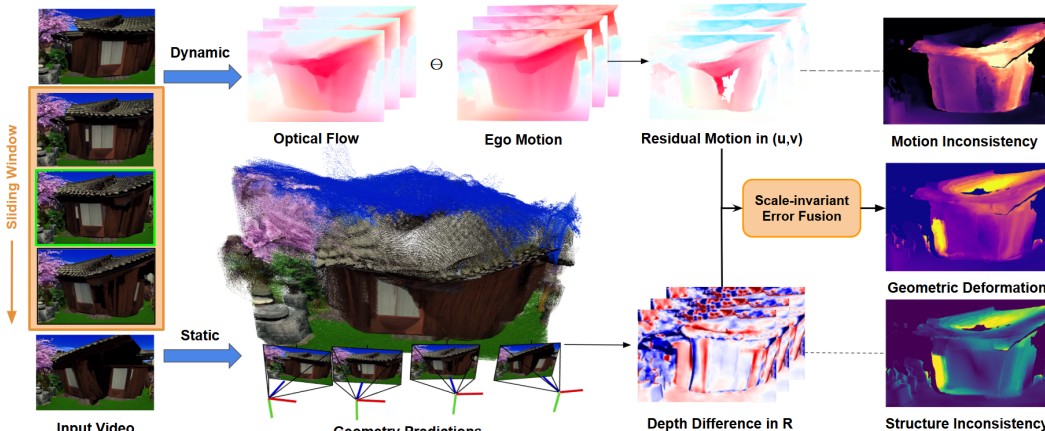

Figure 2: **Geometric deformation detection pipeline.** Our method decomposes inconsistency into two complementary signals. The **dynamic branch** isolates non-rigid motion by computing the **residual motion** (the difference between observed optical flow and camera ego-motion). Concurrently, the **static branch** identifies structural errors by calculating a **depth difference map** from reconstructed 3D geometry. Both error signals are normalized and combined through **scale-invariant fusion** to produce a unified deformation map that precisely localizes geometric artifacts.

## 3 METHODOLOGY

Our goal is to detect and quantify geometric deformation artifacts in generated multi-view videos of static scenes. The core principle is that for a perfectly rigid scene, pixel motion between two frames (optical flow) should be entirely explained by the camera's movement (ego-motion). Any deviation from this rigid motion model indicates a non-rigid deformation. We capture these deviations by decomposing observed inconsistencies into two complementary signals: *motion-based* (dynamic) and *structure-based* (static).

The overall pipeline, illustrated in Figure 2, processes an input video using a sliding-window approach. For each pair of frames, it estimates motion and geometry inconsistencies, normalizes them into a scale-invariant domain, and fuses them into a unified deformation map.

### 3.1 DERIVING MOTION AND GEOMETRIC INCONSISTENCIES

**Motion-Based Inconsistency.** The first signal comes from discrepancies between observed optical flow and the flow predicted by camera motion. We use an optical flow model (Zhang et al., 2025) to compute dense optical flow $F_{t \to t+1}$ between frames $I_t$ and $I_{t+1}$. In parallel, we use a geometry foundation model (Wang et al., 2025) to estimate per-pixel depth $D_t$, camera intrinsics $K_t$, and the relative camera pose $T_{t \to t+1} = [R|t]$. Using these estimates, we construct a **rigid flow field** $F_{\text{rigid}}$ by projecting each pixel $p = (u, v)$ from $I_t$ into $I_{t+1}$:

$$F_{\text{rigid}}(p) = \pi\big(R \cdot (D_t(p)K_t^{-1}\tilde{p}) + t\big) - p,$$

where $\tilde{p} = [u, v, 1]^\top$ is the homogeneous pixel coordinate, and $\pi(\cdot)$ projects 3D points back into 2D using the target camera intrinsics. The difference between observed and rigid flow defines the **residual motion**:

$$F_{\text{residual}}(p) = F_{t \to t+1}(p) - F_{\text{rigid}}(p).$$

This residual highlights non-rigid dynamics, but is valid only for pixels visible in both frames.

**Structure-Based Inconsistency.** To capture inconsistencies in static geometry, including occluded regions where motion is unreliable, we compute a depth reprojection error. The 3D point cloud of $I_t$

**Input Frame** **Warped Frame** **Thin-plate Spline Displacement**

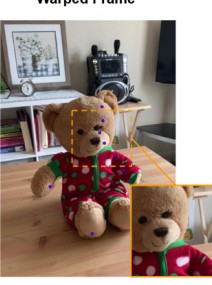 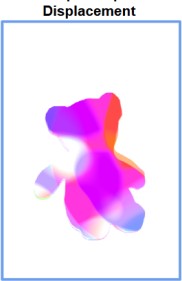

Figure 3: **WARPBENCH generation process.** **(Left)** An input frame with a segmentation mask overlay and sampled source control points (red dots). **(Center)** The warped frame after applying the deformation (inset shows subtle, non-rigid distortion). **(Right)** The dense displacement field from Thin-Plate Spline (TPS) interpolation.

is reprojected into the viewpoint of $I_{t+1}$, yielding a synthetic depth map $D_{t \to t+1}$. This is compared with the independently estimated depth map $D_{t+1}$:

$$\Delta Z(p') = D_{t+1}(p') - D_{t \to t+1}(p'),$$

where $p'$ are pixel coordinates in $I_{t+1}$. The resulting **depth difference map** $\Delta Z$ highlights structural inconsistencies.

## 3.2 SCALE-INVARIANT ERROR FUSION

The residual motion (in pixels) and depth difference (in world units) are not directly comparable. We therefore normalize both into a common, scale-invariant 3D error space using the reference depth $Z = D_t(p)$:

$$(e_x, e_y, e_z) = \left( \frac{\Delta X}{Z}, \frac{\Delta Y}{Z}, \frac{\Delta Z}{Z} \right).$$

Here, $e_z$ is the normalized depth error. Substituting the pinhole camera equations, $\Delta X = Z \cdot \Delta u / f_x$ and $\Delta Y = Z \cdot \Delta v / f_y$, we obtain: $e_x = \frac{\Delta u}{f_x}$, $e_y = \frac{\Delta v}{f_y}$. Thus, motion- and structure-based errors are unified in a depth-invariant domain.

We then fuse these components adaptively. For co-visible pixels, all three terms $(e_x, e_y, e_z)$ are used. For occluded pixels, where residual motion is invalid, we set $(e_x, e_y) = 0$ and rely solely on $e_z$. The final output of our pipeline is the **geometric inconsistency map**, defined as the L2 norm of the active components:

$$M_{\text{geo}}(p) = \sqrt{e_x(p)^2 + e_y(p)^2 + e_z(p)^2}.$$

In addition to this fused map, the **motion inconsistency** and **structure inconsistency** maps are available as intermediate signals, which we use for ablations and diagnostic visualization.

## 4 DATASET

Evaluating geometric artifacts in generated videos is challenging, as artifact regions cannot be reliably annotated. We therefore adopt a two-stage strategy. First, we construct WARPBENCH, a synthetic benchmark that uses Thin Plate Splines (TPS) to mimic localized non-rigid deformations while providing exact ground-truth displacement fields, enabling rigorous validation of our pipeline. Second, to test the realism and robustness of generative models under diverse conditions, we introduce GEO-FLAW, a task-driven benchmark spanning varied scenarios and camera dynamics. Together, these datasets support both controlled validation and realistic evaluation of geometric consistency in video generation.

### 4.1 WARPBENCH: SYNTHETIC DEFORMATION DATASET

**Data Sources.** We instantiate WARPBENCH on two settings: object-centric clips from **CO3D** Reizenstein et al. (2021) (*CO3D-Warp*) and scene-level reconstructions from **ScanNet++** Dai et al. (2017); Yeshwanth et al. (2023) (*ScanNet++-Warp*). In total, WarpBench contains 100 object-centric clips (2,000 frames) and 100 scene-level clips (2,000 frames), spanning 50 object categories and 6 indoor scenes.

**Warp Synthesis.** To simulate non-rigid artifacts, we generate temporally smooth deformations using Thin Plate Splines (TPS). For each clip, we sample $K$ control points from the object mask via farthest-point sampling (FPS) to ensure coverage. Their 2D displacements evolve under a temporal model, and at each frame we fit a TPS to obtain a dense warp. The displacement is spatially localized with a feathered mask, smoothed over time with an exponential moving average (EMA), and applied using differentiable backward sampling.

**TPS Formulation.** Let $C = \{c_i\}_{i=1}^{K}$ be the fixed control points, and $y_{i,t} = c_i + \Delta_{i,t}$ their displaced targets at frame $t$. We fit an affine-plus-RBF mapping $f_t : \mathbb{R}^2 \to \mathbb{R}^2$ with TPS basis $\phi(r) = r^2 \log r$:

$$f_t(x) \ = \ A_t x + a_t + \sum_{i=1}^{K} w_{i,t} \, \phi(\|x - c_i\|),$$

where $A_t \in \mathbb{R}^{2\times 2}$, $a_t \in \mathbb{R}^2$, and $w_{i,t} \in \mathbb{R}^2$. The dense displacement is $U_t(p) = f_t(p) - p$, localized as $\tilde{U}_t(p) = w(p) \, U_t(p)$ with feathered weight $w(p)$. Temporal smoothing gives $\bar{U}_t = \beta \bar{U}_{t-1} + (1-\beta)\tilde{U}_t$. The final warped frame is: $I_t^{\text{def}}(p) = I_t\big(p + \bar{U}_t(p)\big)$. Please refer to Appendix B for a comprehensive list of all parameters used in the WARPBENCH generation pipeline.

**Outputs.** For every warped frame, we release the dense displacement field $\bar{U}_t(p) \in \mathbb{R}^2$ as ground truth, along with its magnitude $M_t(p) = \|\bar{U}_t(p)\|_2$ when a scalar target is needed. These outputs allow precise, per-pixel evaluation of geometric inconsistency detection.

## 4.2 THE GEO-FLAW BENCHMARK

**Benchmark Design.** Deformation artifacts in generated videos appear as temporal geometric inconsistencies To systematically evaluate this, we introduce **Geo-Flaw**, a small benchmark inspired by core 3D vision tasks where structural consistency is essential. It spans both commercial and open-source models and covers four categories: *object-centric reconstruction*, *indoor navigation*, *large-scale outdoor reconstruction*, and a stress-testing case targeting *challenging surfaces and edges*.

Within each category, we generate videos under two camera regimes: *slow, smooth motion*, which exposes subtle "breathing" or "melting" artifacts, and *fast, dynamic motion*, which stresses multi-view coherence under aggressive perspective changes. This design enables structured evaluation of a model's ability to produce stable and plausible 3D geometry. For a full summary of the benchmark's composition, including the scenarios detailed in Table 3, please see Appendix A.

**Model Selection.** For each open-source model, we generate clips per scenario, resulting in ¿50 videos per model. For the commercial system Sora, we follow its released content and obtain 80 object-centric, 96 indoor navigation, 53 large-scale outdoor, and 80 challenging-surface clips (309 videos in total). Our evaluation therefore spans both leading commercial systems and recent open-source state-of-the-art models, including WAN 2.2 (Wan et al., 2025) and CogVideoX (Hong et al., 2022; Yang et al., 2024).

# 5 EXPERIMENT

**Experiment Design.** Our experiments are structured to evaluate both the capability of our method and its utility for studying generative models. We first use WARPBENCH to validate that our pipeline can detect anomalous frames and localize spatial deformations under controlled, ground-truth conditions. We then turn to GEO-FLAW, where our method serves as a diagnostic tool for benchmarking commercial and open-source video generation models across diverse scenarios without ground-truth annotations.

## 5.1 PIPELINE EVALUATION ON SYNTHETIC DATA

We evaluate our method through two complementary tasks: single-frame anomaly detection (temporal) and pairwise spatial localization (spatial). These experiments are conducted on the **CO3D-Warp** and **Scannet-Warp** datasets and compared against baseline methods.

Table 1: **Pairwise spatial localization and correlation results.** Higher is better for all metrics. "Structure Only" uses depth reprojection error alone, "Motion Only" uses residual motion, "Fusion (Full)" combines both in a scale-invariant domain, and "Fusion (Occlusion-Aware)" adds depth only in occluded regions. MEt3R serves as a baseline.

| Method | CO3D-Warp | | | Scannet-Warp | | |
|---|---|---|---|---|---|---|
| | AP (%) ↑ | IoU (%) ↑ | SRCC ↑ | AP (%) ↑ | IoU (%) ↑ | SRCC ↑ |
| MEt3R (baseline) | 16.26 | 15.95 | -0.176 | 30.34 | 33.13 | -0.351 |
| Structure Only | 25.64 | 3.20 | 0.079 | 51.71 | 14.11 | 0.183 |
| Motion Only | **64.90** | **44.48** | **0.581** | **87.12** | **52.36** | **0.706** |
| Fused (Full) | 60.63 | 41.69 | 0.554 | 82.70 | 48.87 | 0.547 |
| Fused (Occlusion-Aware) | 63.49 | 43.42 | 0.561 | 83.66 | 49.20 | 0.557 |

**Experiment 1: Single-Frame Anomaly Detection.** In this task, a single frame within a 10-frame clip is randomly replaced by its warped version, and the goal is to identify the anomalous frame. Performance is measured using **Detection Accuracy (%)**, i.e., the fraction of clips where the manipulated frame is correctly identified.

**Analysis.** Results in Table 4 show that motion is the most reliable cue for anomaly detection: *Motion Only* achieves the best accuracy on CO3D-Warp (71.59%) and strong results on ScanNet-Warp (89.23%). Fusion with depth is mixed—"Fusion (Full)" gives the highest accuracy on ScanNet-Warp (92.31%) but lags behind motion alone on CO3D-Warp, suggesting depth can introduce noise in less reliable settings. The "Fusion (Occlusion-Aware)" ablation, which applies depth only in occluded regions, performs closer to Motion Only, showing the benefit of targeted depth integration. The baseline (MEt3R) performs poorly, underscoring the challenge of the task and the advantage of explicitly modeling motion and geometry.'

**Experiment 2: Pairwise Spatial Localization.** This experiment evaluates the ability of the method to produce spatial maps of manipulations. Given a pair of frames (real vs. warped), the system generates an inconsistency map that is compared against the ground truth. We report **AP (%)**, **IoU (%)**, and **Spearman's Rank Correlation (SRCC)**[1] to jointly assess localization precision, overlap with ground-truth masks, and consistency with manipulation intensity.

**Analysis.** As shown in Table 1, motion is the dominant cue for spatial localization: *Motion Only* outperforms all variants, reaching 87.12% AP, 52.36% IoU, and 0.706 SRCC on ScanNet-Warp. Fusion with depth slightly reduces performance, suggesting conflicts in noisy regions,

Figure 4: **Single-frame anomaly detection accuracy (%).** Higher is better. "Structure Only" uses depth reprojection error alone, "Motion Only" uses residual motion from optical flow, "Fusion (Full)" combines both in a scale-invariant domain, and "Fusion (Occlusion-Aware)" applies depth error only in occluded regions. MEt3R serves as a baseline.

| Method | CO3D-warp | Scannet-Warp |
|---|---|---|
| MEt3R (baseline) | 6.82 | 15.38 |
| Structure Only | 42.05 | 84.62 |
| Motion Only | **71.59** | 89.23 |
| Fused (Full) | 55.68 | **92.31** |
| Fused (Occlusion-Aware) | 52.27 | 87.69 |

though the "Fusion (Occlusion-Aware)" variant remains close to Motion Only, highlighting the benefit of targeted depth use. Negative SRCC values for MEt3R indicate not just failure but anti-correlation with ground truth, underscoring the challenge of this task and the effectiveness of our approach.

## 5.2 BENCHMARKING GENERATIVE MODELS WITH OUR PIPELINE

Having validated our pipeline on synthetic data, we now apply it to its primary domain: analyzing videos from generative models. In this context, where ground-truth masks are unavailable, our

---

[1]Spearman's Rank Correlation is a non-parametric measure of rank correlation that evaluates the strength and direction of a monotonic relationship between two ranked variables, defined as $\rho = 1 - \frac{6\sum d_i^2}{n(n^2-1)}$, where $d_i$ is the difference between the ranks of paired observations and $n$ is the number of pairs.

Table 2: **Video generation evaluation across scenarios.** Metrics shown are Motion Inconsistency, Structure Inconsistency, and the final deformation scores with and without occlusion-aware fusion. Each score is the mean value of its corresponding inconsistency map, averaged over the video. Lower values indicate fewer geometric artifacts.

| Type | Model | Motion ↓ | Structure ↓ | Fused (Occ) ↓ | Fused (Full) ↓ |
|---|---|---|---|---|---|
| **Object-centric** | | | | | |
| Commercial | Sora | **0.018749** | **0.106039** | **0.169207** | **0.186912** |
| Open-source | WAN 2.2 | 0.033315 | 0.354326 | 0.245789 | 0.262989 |
| Open-source | CogVideoX | 0.024880 | 0.165006 | 0.311831 | 0.367868 |
| **Indoor navigation** | | | | | |
| Commercial | Sora | **0.026924** | **0.182063** | **0.138727** | **0.149884** |
| Open-source | WAN 2.2 | 0.052337 | 0.479448 | 0.147014 | 0.157955 |
| Open-source | CogVideoX | 0.032981 | 0.247008 | 0.254227 | 0.306632 |
| **Outdoor reconstruction** | | | | | |
| Commercial | Sora | 0.022506 | 0.188681 | **0.148556** | **0.161185** |
| Open-source | WAN 2.2 | **0.015693** | 0.385315 | 0.152012 | 0.162418 |
| Open-source | CogVideoX | 0.028539 | **0.140516** | 0.335744 | 0.391256 |
| **Challenging (stress test)** | | | | | |
| Commercial | Sora | 0.033955 | 0.408964 | **0.156814** | **0.171525** |
| Open-source | WAN 2.2 | **0.032849** | 0.443181 | 0.189504 | 0.206616 |
| Open-source | CogVideoX | 0.033371 | **0.094839** | 0.353770 | 0.452582 |

method serves as a diagnostic tool to quantitatively score and qualitatively assess violations of 3D geometric consistency.

**Quantitative Benchmark.** We first apply our pipeline to score videos from several leading generative models—Sora, WAN 2.2, and CogVideoX—across a variety of scenarios. A lower inconsistency score, as measured by our method, indicates stronger geometric stability and fewer deformation artifacts. The aggregated results of this benchmark are summarized in Table 2.

The quantitative results reveal clear performance differences among the models. Sora consistently achieves the lowest inconsistency scores across all scenarios, indicating a higher degree of geometric stability. WAN 2.2 performs competitively, particularly on outdoor scenes, but is less robust indoors. CogVideoX exhibits the highest fused errors, especially in challenging cases. Notably, across all models, the occlusion-aware fusion variant typically yields lower error scores than the full fusion, reinforcing the benefit of selectively integrating depth cues when analyzing generated content.

**Qualitative Analysis.** Qualitative examples provide visual intuition for these quantitative scores and highlight the types of errors our method detects. A common failure mode, even for top-performing models, is the inability to maintain the rigidity of simple rotating objects. Figure 6 illustrates this with a deforming globe, where our method effectively captures the spurious motion and structural warping that contribute to a higher inconsistency score.

Furthermore, the interpretability of our inconsistency maps is crucial for their utility as a diagnostic tool. In Figure 5, we compare our method against baselines designed for generic non-rigid motion detection. On a generated video with subtle artifacts, methods like SegAnyMo and MEt3R either fail to detect the localized motion or produce blurred, unspecific score maps that mask the entire object. In contrast, our pipeline generates precise, interpretable maps that isolate the specific areas of geometric distortion, offering more actionable feedback on model performance.

## 5.3 FINDINGS ON VIDEO GENERATION MODELS

*"And Then There Were None."* Our most challenging geometric consistency tests revealed a universal vulnerability: all evaluated models failed to maintain structural integrity. This highlights a common point of failure in current video generation methods, especially under stress tests targeting fine patterns, reflections, refractions, and dense edges that are prone to artifacts.

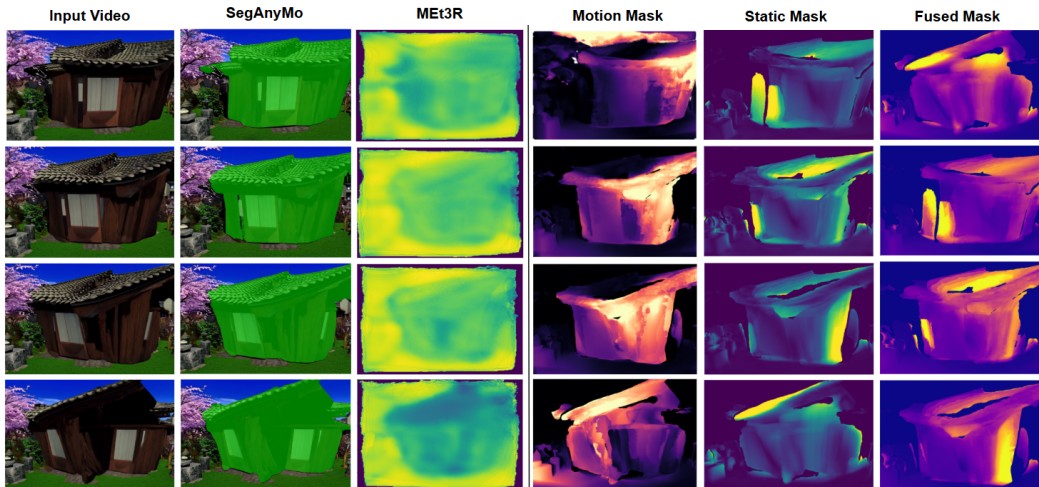

Figure 5: Qualitative comparison on deformation artifacts. Top row: input video with geometric deformation. SegAnyMo fails to predict localized motion, while MEt3R produces a blurred score map without region-level detail. In contrast, our method produces interpretable maps that highlight subtle, localized geometric distortions rather than masking entire objects.

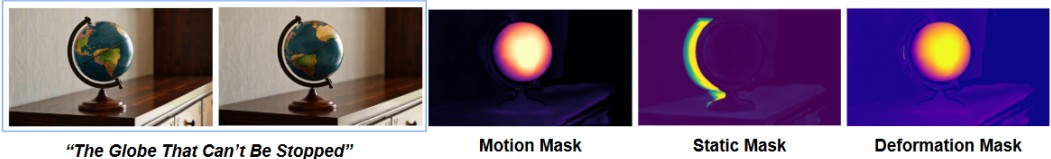

Figure 6: **The Globe That Cannot Be Stopped**: Even state-of-the-art video generation models struggle to render a rigid globe without introducing spurious motion and deformation, as revealed by our motion, static, and fused inconsistency maps.

*"The Globe That Can't Be Stopped."* We also identified a consistent and surprising failure mode: the inability of all models to generate a simple, rigidly rotating globe. Instead of producing stable motion, models introduce subtle non-rigid deformations or irregular rotations. A possible reason is bias in training data, where most examples of globes appear in motion, leading models to conflate object persistence with deformation or drift.

# 6 CONCLUSION

In this work, we introduced a novel pipeline for detecting and quantifying geometric deformation artifacts in videos of static scenes. Our central finding is that **residual motion is the most potent indicator of deformation**, consistently and significantly outperforming cues derived from scene structure, such as depth. We demonstrated that a focused motion-based analysis is paramount, as naively fusing depth information can often dilute the primary signal.

By deploying our method as a metric, we benchmarked leading generative models and uncovered several important findings. We identified universal vulnerabilities, with all models failing challenging "stress tests," and consistent failure modes in seemingly simple scenarios, such as rendering a rigidly rotating globe. These results not only validate our pipeline as an effective diagnostic tool but also highlight that maintaining geometric consistency remains a critical challenge for even state-of-the-art video generation models.

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

## A    THE GEO-FLAW BENCHMARK COMPOSITION

Table 3: Composition of the Geo-Flaw benchmark. Our dataset is structured around four evaluation scenarios designed to probe geometric consistency, each tested with both slow and fast camera dynamics.

| Evaluation Scenario | Dynamics | Example Scenarios |
|---|---|---|
| **Object-Centric Reconstruction** | **Slow** | 360° orbit of globe; upward pan of statue; push-in on clay pot. |
| *Objective:* Maintain rigid, detailed geometry of a single isolated object. | **Fast** | Rapid orbit of sports car; zig-zag approach to dollhouse; armor hall fly-through. |
| **Indoor Navigation** | **Slow** | Library dolly; gallery arc; cathedral glide. |
| *Objective:* Preserve coherent room layout and global structure during traversal. | **Fast** | Hallway sprint; server-room zig-zag; staircase swoop. |
| **Outdoor Large-Scene Reconstruction** | **Slow** | Mountain sweep; ruins glide; forest track. |
| *Objective:* Ensure consistency across expansive environments with layered depth. | **Fast** | Rooftop traverse; amusement-park fly-through; refinery sweep. |
| **Challenging Surfaces & Edges** | **Slow** | Mosaic macro track; chrome-engine orbit; chandelier orbit. |
| *Objective:* Stress-test fine patterns, reflections, refractions, and dense edges prone to artifacts. | **Fast** | Grand staircase ascent; bookcase fly-through; glass corridor traverse. |

## B    WARPBENCH GENERATION PARAMETERS

Here we detail the parameters used in the WARPBENCH data generation pipeline, described in Section 4.1.

**Control Points.** For each video clip, we sample a fixed set of $K = 24$ control points from the initial frame's segmentation mask using farthest-point sampling to ensure broad spatial coverage. These points remain fixed for the duration of the clip to provide a stable basis for deformation.

**Temporal Motion Model.** The per-frame offsets for the control points, $\Delta_{i,t}$, are generated using an AR(1) autoregressive process to ensure temporally smooth yet non-trivial motion. The update rule is:

$$\Delta_{i,t} = \rho\Delta_{i,t-1} + \sigma\epsilon_t,$$

where $\epsilon_t \sim \mathcal{N}(0, I)$ is random Gaussian noise. We use a high correlation coefficient $\rho = 0.95$ to ensure smoothness and a noise standard deviation of $\sigma = 0.6$ to introduce variation.

**Deformation Magnitude.** After the raw displacement field $U_t$ is generated, it is rescaled to match a predefined target magnitude. This provides direct control over the deformation strength. For our experiments, the target magnitude (the mean per-pixel displacement within the mask) is sampled uniformly for each clip from a range of $[3, 8]$ pixels.

**Mask Localization and Feathering.** To ensure the warp is localized to the object of interest and blends smoothly with the background, we modulate the displacement field with a weight map $w(p)$. This map is derived from the ground-truth segmentation mask by first eroding the mask by 10 pixels and then applying a cosine falloff over a 20-pixel "feathering" band at the edge. This creates a soft transition from the fully warped region to the static background.

**Temporal Smoothing (EMA).** As a final step to prevent unnaturally jerky motion, we apply an Exponential Moving Average (EMA) to the sequence of displacement fields. We use a smoothing factor of $\beta = 0.8$ in the update rule $\bar{U}_t = \beta\,\bar{U}_{t-1} + (1-\beta)\tilde{U}_t$.

**Data packaging.**    Each sample is the tuple $\left(I_t,\, I_t^{\text{def}},\, U_t,\, M_t\right)$.    Images $I_t$, $I_t^{\text{def}}$: PNG (8-bit sRGB). Displacement $U_t$: float32 array $(H{\times}W{\times}2)$ in pixel units.    Magnitude

$M_t$: float32 ($H \times W$). A JSON manifest accompanies each clip with hyperparameters $(K, s, \rho, \sigma, \lambda, \beta,$ feather radius, clamps), PRNG seeds, and generation flags.

**Defaults used in our experiments.** Unless otherwise noted, we use: $T=20$; control-point density targeting $\delta$ points per $10^4$ mask pixels with bounds $20 \leq K \leq 80$ and minimum spacing $s$ px; AR(1) coefficient $\rho$; target RMS amplitude $A$ (as a fraction of the short image side) with $\sigma = \sqrt{1 - \rho^2}\, A$; displacement clamp $A_{\max}$; TPS regularization $\lambda$ (on normalized coordinates); feather radius $r_f$; EMA coefficient $\beta$. Exact values and ranges are reported in the released configs.

**Stability safeguards.** We reject and resample a clip if TPS induces excessive distortion (e.g., more than a small fraction of pixels with $|\det J_{f_t}(p)| < \tau_J$) or if median $\|U_t\|$ exceeds a bound. We also cap $\|\Delta_{i,t}\| \leq A_{\max}$ during AR(1) generation.

