# OpenReview forum: "Detecting Geometric Deformation in Visual Generation"
_ICLR.cc/2026/Conference — ICLR 2026 Conference Withdrawn Submission_

### Official Review · Reviewer_449i · 2025-10-25

**Soundness:** 2
**Presentation:** 3
**Contribution:** 2
**Rating:** 4
**Confidence:** 4

**Summary:**

This paper studies geometric inconsistencies in text-to-video and multi-view generation models. The authors propose a geometry-based detection system that compares the observed optical flow with the rigid flow predicted from estimated depth and camera motion, to find non-rigid deformations. The method combines motion and depth cues in a scale-invariant and occlusion-aware way to produce pixel-level deformation maps. To evaluate the approach, the authors create two benchmarks. Experiments show that their method performs better than feature-based metrics like MEt3R and coordinate-only methods like WorldScore. It also provides more accurate localization and clearer interpretations. Finally, the results show that even state-of-the-art models become structurally unstable under stress tests.

**Strengths:**

1. The paper presents a simple, training-free, geometry-based method that directly measures rigidity errors using residual motion and depth reprojection.
2. The per-pixel deformation maps are easy to understand and helpful for diagnosing geometric issues, offering clear visual explanations.
3. WARPBENCH and Geo-Flaw together enable both controlled quantitative testing and realistic evaluation.
4. The framework is flexible — it can also be used for moving-object segmentation, showing its broader applicability.

**Weaknesses:**

1. The method depends on pretrained optical flow, depth, and camera pose estimators. Errors in these components could carry over and distort the deformation maps.
2. The paper should include an analysis of how sensitive the method is to the quality of flow and depth estimation.
3. The results suggest that simple depth fusion can reduce performance, but the paper does not fully explain how the occlusion-aware fusion works or what thresholds are used.
4. The pipeline includes several estimation steps (flow, depth, pose), which may make real-time use difficult. Also, in my view, this work mainly combines existing tools. It feels more like an engineering solution than a novel scientific contribution.
5. The Geo-Flaw benchmark includes commercial videos (such as Sora) without exact ground truth, so the fairness and reproducibility of those comparisons could be discussed more clearly.
6. The fusion step is described as “scale-invariant,” but if the depth estimates are biased or unstable, the final error map could still be inaccurate.

**Questions:**

1. How sensitive is the method to errors in depth and pose estimation, especially in areas that are textureless or reflective?
2. Could the fusion weights between motion and depth errors be learned adaptively instead of being fixed by hand?
3. How does the method handle scenes with real moving objects? Does it mistake them for deformations, or could it be extended to tell real motion apart from artifacts?
4. The method assumes that all non-rigid motion means geometric deformation. But in generated videos, lighting changes or texture flicker can also cause optical flow differences. How do you distinguish real geometric errors from these non-geometric effects?
5. The approach assumes a fully static scene, but in reality, small natural motions (like rippling water or moving leaves) are common. Would the method incorrectly flag these as artifacts?
6. The method is based on a perfect rigid 3D projection model, but modern generative videos often have perspective shifts or stylized camera effects. Does your rigid model still apply in those cases?

---

### Official Review · Reviewer_2rBk · 2025-11-01

**Soundness:** 1
**Presentation:** 2
**Contribution:** 2
**Rating:** 4
**Confidence:** 4

**Summary:**

This paper tries to detect the geometry inconsistency for generated videos from video generation models, which is inspired by the non-rigid artifacts observed from current generated videos. To enable a quantitative study of this kind of failure model for video generation, this paper repurpose standard 3D datasets like CO3D and ScanNet++ and warp them to mimic the artifacts. On this benchmark, the paper evaluates several state of the art models and highlights the remained difficulties for future inspiration.

**Strengths:**

1) The benchmark is strongly motivated by observed failure modes in state-of-the-art video generation models.
2) The paper introduces a benchmark for quantitatively analyzing geometric artifacts, notably employing thin-plate splines (TPS) to synthesize non-rigid deformations.
3) The paper is easy to follow.

**Weaknesses:**

Overall, I believe this paper addresses a notable failure case  for video generation models. However, for the benchmark setup, experiments and presentation, I still have some major concerns.
1) The connection between the stated motivation (real-world generative artifacts) and the proposed benchmark is tenuous on two fronts:
•	The benchmark approximates artifacts using thin-plate splines (TPS). This synthetic transformation may not capture the full complexity and spectrum of geometric failures observed in actual generated videos.
•	Geometric consistency is quantified via an external motion model. As this motion model is inherently imperfect, it is difficult to deconfound its own estimation errors from the 3D geometry errors of the video being evaluated. This ambiguity calls into question the reliability of the final quantification.
2) The benchmark's synthetic nature may introduce a significant domain gap. Real-world 3D consistency is often closely tied to object semantics and contextual understanding. It is unclear whether a model's performance on this synthetic benchmark will correlate with its ability to maintain consistency in real, semantically complex generated videos.
3) The empirical results are difficult to interpret and appear contradictory in places.
•	The paper does not adequately explain why the "fused model" performs slightly worse than the "motion only" model in the pairwise localization results. This is counter-intuitive and suggests the "structure cue" (depth projection) may be unreliable or detrimental.
•	Furthermore, the results seem to diverge between Table 1 and Table 2 (which is incorrectly labeled as "Figure 4" in the text). The relative performance of the "full model" versus the "motion only" model changes across datasets, which obscures the paper's central claims and makes the conclusions less convincing.
4)  The qualitative example in Figure 6 is a poor illustration. The scene itself (e.g., a person speaking) clearly violates the method's underlying assumption of a rigid 3D environment. As such, it fails to demonstrate the specific non-rigid artifact failure mode the benchmark is designed to detect.
5)  The paper's quality is diminished by a lack of careful proofreading and several conspicuous typos. For example:
•	The text repeatedly references "Figure 4" (e.g., in the caption for Table 2) when it should be "Table 2."
•	There are formatting errors with quotation marks (e.g., ” instead of “ in lines 429 and 467).
•	Dataset naming is inconsistent (e.g., "CO3D-warp" vs. "CO3D-Warp").

**Questions:**

Please see the weaknesses.

---

### Official Review · Reviewer_Z1zP · 2025-11-01

**Soundness:** 2
**Presentation:** 2
**Contribution:** 2
**Rating:** 2
**Confidence:** 4

**Summary:**

This paper focus on a key limitation in text-to-video and multi-view generation models, that these models often produce geometric deformation artifacts across viewpoints. To address this, the authors propose a geometry-grounded detection pipeline that fuses residual motion and depth reprojection errors into interpretable, dense deformation maps. To verify the performance of the pipeline, the authors introduce a synthetic dataset with dense ground-truth warps, and a comprehensive benchmark and analysis of geometric artifacts.

**Strengths:**

1. The paper focuses on 3D geometric integrity in video generation, which is an underexplored yet crucial issue.
2. The approach sounds feasible.
3. The work is notable for its training-free design.
4. The authors conduct complete experiments, and the proposed benchmarks are well designed.

**Weaknesses:**

1. The approach assumes static videos with no moving objects, which limits applicability to dynamic scenes.
2. The paper focuses only on detecting geometric artifacts, but does not discuss how the proposed method could help to correct these artifacts in generative videos.
3. The dataset uses thin-plate-spline warps to create controlled artifacts, which may not capture the full variety of geometric deformations found in real generative models.
4. The method works pairwise between frames, but multi-frame temporal consistency could bring cumulative geometric drifts, which is not explored.

**Questions:**

1. How robust is the proposed method with mild dynamics in the video, such as small or slow moving objects?
2. How sensitive is the pipeline to errors in depth or camera pose estimation?

---

### Official Review · Reviewer_ZzUV · 2025-11-01

**Soundness:** 3
**Presentation:** 3
**Contribution:** 3
**Rating:** 4
**Confidence:** 3

**Summary:**

This paper proposes a new pipeline for detecting and quantifying geometric deformation artifacts—such as "melting" or "breathing"—in videos generated by models, specifically focusing on the static scene regime where the camera moves but objects should remain rigid. The method operates by comparing the observed optical flow between frames with a predicted rigid flow field derived from estimated camera motion and scene depth. The difference, or "residual motion," is combined with a depth reprojection error in a scale-invariant manner to produce a dense, per-pixel deformation map that is robust to occlusions. To evaluate this, the authors introduce two new benchmarks: WARPBENCH, a synthetic dataset with ground-truth thin-plate-spline warps, and Geo-Flaw, a task-oriented benchmark for evaluating generative models. Experiments show the method can localize these artifacts and outperforms baselines like MEt3R, and it is used to diagnose issues in models like Sora and WAN 2.2.

**Strengths:**

1. The paper proposes a novel, training-free pipeline for detecting geometric artifacts, which is a valuable diagnostic tool.

2. The introduction of WARPBENCH is a strong contribution, providing a synthetic dataset with dense, ground-truth warp fields for rigorous, quantitative evaluation, which was previously lacking.

3. The experimental evaluation is thorough, validating the method on synthetic data and then applying it to benchmark state-of-the-art generative models like Sora on the new Geo-Flaw dataset.

4. The method's ability to produce dense, interpretable deformation maps is a significant advantage over single-score metrics (like WorldScore) or diffuse feature-based maps (like MEt3R).

**Weaknesses:**

1. The primary weakness is the limited significance and niche scope of the problem. The paper focuses only on static scenes, which is a simplification of the general video generation task. This specific "melting" artifact is just one of many failure modes, and it's unclear if this warrants such a specialized tool, especially as future models might solve this artifact implicitly.

2. The method is fundamentally dependent on the performance of two large, pre-trained foundation models: an optical flow model (UFM) and a geometry foundation model (Vggt). The paper's pipeline is effectively a post-processing step on their outputs. Any failure in these upstream models (e.g., poor depth estimation, or flow failure on large viewpoint changes) will cascade and cause the pipeline to fail, but this dependency is not explored.

3. The paper lacks a sufficient ablation study on the key hyperparameters of the pipeline itself or the dataset generation. For instance, the WARPBENCH generation depends on many parameters ($K$, $\rho$, $\sigma$, $\beta$, etc.), and it is unclear how their settings impact the evaluation.

4. The paper's own results and conclusion somewhat marginalize its fusion contribution. Table 1 and the analysis show that "Motion Only" consistently outperforms the fused methods. The conclusion reinforces this, stating "residual motion is the most potent indicator" and that "naively fusing depth... can often dilute the primary signal". This makes the complex scale-invariant fusion seem unnecessary or even detrimental, which is a confusing message.

**Questions:**

1. The method's performance hinges on the quality of the optical flow and geometry foundation models. How does the pipeline's performance change if these specific models are replaced with alternatives (e.g., a different depth estimator or a classical flow algorithm)? How much of the strong result is due to the power of UFM and Vggt versus the logic of the pipeline itself?

2. The paper focuses exclusively on static scenes. Could the core principle be extended to dynamic scenes? How would the pipeline differentiate between plausible non-rigid motion (e.g., a person waving) and artifactual non-rigid motion (e.g., a person's arm "melting")?

3. Given that the "Motion Only" ablation outperforms the "Fused (Occlusion-Aware)" method in all metrics on Table 1 (AP, IoU, and SRCC), why is the fusion component still presented as a core, positive contribution of the pipeline? This seems contradictory, as the data suggests the depth/structure component only adds noise.

---

### Note · Authors · 2025-11-12

I have read and agree with the venue's withdrawal policy on behalf of myself and my co-authors.